# Identification of a sugarcane bacilliform virus promoter that is activated by drought stress in plants
Sheng-Ren Sun[1,2,3], Xiao-Bin Wu[4], Jian-Sheng Chen[5], Mei-Ting Huang[1], Hua-Ying Fu[1], Qin-Nan Wang[2], Philippe Rott [6,7] ✉ & San-Ji Gao[1] ✉

Sugarcane (*Saccharum* spp.) is an important sugar and biofuel crop in the world. It is frequently subjected to drought stress, thus causing considerable economic losses. Transgenic technology is an effective breeding approach to improve sugarcane tolerance to drought using drought-inducible promoter(s) to activate drought-resistance gene(s). In this study, six different promoters were cloned from sugarcane bacilliform virus (SCBV) genotypes exhibiting high genetic diversity. In β-glucuronidase (GUS) assays, expression of one of these promoters ($P_{SCBV-YZ2060}$) is similar to the one driven by the CaMV 35S promoter and >90% higher compared to the other cloned promoters and Ubi1. Three SCBV promoters ($P_{SCBV-YZ2060}$, $P_{SCBV-TX}$, and $P_{SCBV-CHN2}$) function as drought-induced promoters in transgenic *Arabidopsis* plants. In *Arabidopsis*, GUS activity driven by promoter $P_{SCBV-YZ2060}$ is also upregulated by abscisic acid (ABA) and is 2.2–5.5-fold higher when compared to the same activity of two plant native promoters ($P_{ScRD29A}$ from sugarcane and $P_{AtRD29A}$ from *Arabidopsis*). Mutation analysis revealed that a putative promoter region 1 (PPR1) and two ABA response elements (ABREs) are required in promoter $P_{SCBV-YZ2060}$ to confer drought stress response and ABA induction. Yeast one-hybrid and electrophoretic mobility shift assays uncovered that transcription factors ScbZIP72 from sugarcane and AREB1 from *Arabidopsis* bind with two ABREs of promoter $P_{SCBV-YZ2060}$. After ABA treatment or drought stress, the expression levels of endogenous ScbZIP72 and heterologous GUS are significantly increased in $P_{SCBV-YZ2060}$:GUS transgenic sugarcane plants. Consequently, promoter $P_{SCBV-YZ2060}$ is a possible alternative promoter for genetic engineering of drought-resistant transgenic crops such as sugarcane.

Crop growth and development are constantly exposed to environmental conditions, including biotic and abiotic stresses that threaten world's food security[1]. In global climate change, drought is one of the major environmental stresses affecting crop growth and productivity[2], including plant resistance to diseases[3]. Sugarcane (*Saccharum* spp. hybrids) is an important sugar and bioenergy crop, accounting for about 80% and 40% of sugar and ethanol productions worldwide, respectively[4]. Sugarcane is grown in tropical and sub-tropical climate areas where drought stress can affect plant germination, grand growth, and maturation[5]. Enhancing drought tolerance

to improve sugarcane productivity has become an essential challenge to meet the sugar consumption of expanding population worldwide. An effective approach to reach that goal is to develop transgenic plants with overexpressed drought-resistance genes or knocked-down drought-susceptibility genes[5,6].

The promoter is a key element for proper functioning of any transgene[7]. Diverse constitutive promoters have been used in crop genetic improvement via transgenics. These include viral promoters such as the *35S* from cauliflower mosaic virus (CaMV) of the virus family *Caulimoviridae*[8],

[1]National Engineering Research Center for Sugarcane, Fujian Agriculture and Forestry University, Fuzhou 350002 Fujian, China. [2]Institute of Nanfan & Seed Industry, Guangdong Academy of Sciences, Guangzhou 510316 Guangdong, China. [3]Hainan Yazhou Bay Seed Laboratory, Sanya 572024 Hainan, China. [4]College of Ocean and Earth Sciences, Xiamen University, Xiamen 361000 Fujian, China. [5]College of Life Sciences, Fujian Agriculture and Forestry University, Fuzhou 350002 Fujian, China. [6]CIRAD, UMR PHIM, 34398 Montpellier, France. [7]PHIM Plant Health Institute, Univ Montpellier, CIRAD, INRAE, Institut Agro, IRD, Montpellier, France. ✉e-mail: philippe.rott@cirad.fr; gaosanji@fafu.edu.cn

and native promoters from higher plants such as *Ubiquitin 1* (*Ubi1*) from maize[6,9]. Strong constitutive expression of foreign genes may, however, have multiple or negative pleiotropic effects in transgenic plants[10]. Constant expression of foreign genes in untargeted tissues and at inappropriate developmental time usually leads to energy waste, abnormal morphology, and delayed development[10,11]. Such negative effects in transgenic plants can be reduced or eliminated using different genetic background promoters or by alteration of the expression level of the targeted genes[12]. Similarly, the constitutive genome editing such as the clustered regularly interspaced palindromic repeat-Cas system (CRISPR-Cas) of some vital genes can also result in pleiotropic effects in multiple tissues or for other genes. Consequently, tissue-specific or inducible CRISPR-Cas genome editing is a desired approach to reduce the off target and pleotropic effects in gene editing of crop plants[13].

Sugarcane bacilliform viruses (SCBV; genus *Badnavirus*, family *Caulimoviridae*) that are associated with leaf fleck in sugarcane possess circular, double-stranded DNA genomes encoding three open reading frames (ORFs)[14,15]. These viruses show high genetic diversity and 25 different genotypes (SCBV-A to SCBV-Y) have been reported so far[16]. In the family *Caulimoviridae*, the intergenic region of the viral genome is a potential promoter region (PPR)[9,17]. Several SCBV promoters drive heterologous and strong gene expression resulting in constitutive or tissue-specific patterns in monocots and dicots[9,18–21]. Among those, the promoter from sugarcane bacilliform MO virus (SCBMOV-MOR) was first reported to be a constitutive promoter driving reporter gene expression in plants[18,19], whereas tissue specificity of this promoter was observed in different transgenic crops[20]. Another promoter from sugarcane bacilliform IM virus (SCBIMV-QLD) conferred highest activity in meristems in transgenic sugarcane[21], and the enhancer of this promoter is a *cis*-acting element activating gene transcription[22]. Additionally, promoter SCBV21 of isolate SCBV-TX from Texas (USA) is a tissue-regulated promoter with preferential expression in the culm vascular bundles[9]. However, SCBV stress-induced promoters have not been reported so far.

Abscisic acid (ABA) plays a critical role in coordinating the response of plants to various environmental changes[23]. In response to drought, plants use ABA-mediated pathways to control water loss by closing stomata and adjusting morphological and physiological traits. These traits include increased root growth, reduced stem and leaf growth, regulation of stress-response genes, and antioxidant production to detoxify reactive oxygen species[23–25]. Promoters of the ABA-response genes possess a *cis*-acting element (ABRE, PyACGTGG/TC) that is essential for the response to ABA[26]. The ABREs in ABRE-binding (AREB) proteins or ABRE-binding factors (ABFs) form a group of basic region/leucine zipper motif (bZIP) transcription factors (TFs) from *Arabidopsis* that are key regulators in mediating ABA-triggered plant tolerance to abiotic stresses[27,28]. A series of bZIP TFs specifically bind to the ABRE elements in promoters of ABA-responsive genes, thus activating their transcript expression[29,30]. In higher plants, the bZIP proteins form a large family of TFs that are involved in stress-related responses in addition to plant development[30,31]. For instance, over-expression of genes *OsbZIP42* and *OsbZIP72* increased the tolerance to abiotic stresses in transgenic rice[32,33]. Similarly, genes *ZmbZIP72* and *ZmbZIP4* from maize overexpressed in *Arabidopsis* conferred tolerance to salt and drought[34,35].

In transgenic plants, the use of plant virus promoters rather than native plant promoters is a delicate strategy because transgenes sharing homology in their promoter regions are likely inactivated by transcriptional gene silencing[36,37]. The use of alternative promoters is essential to minimize the risk of transgene silencing in multigene transformation[38,39]. Efficient transgene silencing is a crucial limitation to sugarcane improvement by genetic transformation[6,40]. In addition, to counteract the negative impact of plant native promoters in modulating gene expression, artificial synthetic promoters are accurate, smart, and versatile for driving gene expression and consequently enhancing desirable traits in crops[41]. Therefore, we carried out an investigation into drought-responsive promoters from different isolates

of SCBV. Activity of these promoters was determined based on β-glucuronidase (*GUS*) gene expression in transgenic *Arabidopsis* exposed to drought-stress conditions. Subsequently, regulation of SCBV promoter $P_{SCBV-YZ2060}$ and its critical *cis*-acting elements by ScZIP72 from sugarcane was characterized in transgenic *Arabidopsis* and sugarcane in response to drought stress. Data obtained in this study highlight the possible use of an alternative, drought-inducible, viral promoter for genetic improvement of sugarcane and other crops.

## Results

### Sequence comparison of eight SCBV promoters

To explore the genetic diversity and characteristics of different SCBV promoters, six SCBV partial genomic fragments of approximately 3.0 kb were amplified and sequenced in this study. Additionally, genomic fragments from published isolates SCBMOV-MOR and SCBV-TX were also obtained and sequenced. All eight fragments contained a partial gene coding for the RT/RNase H and the entire promoter region. A phylogenetic tree constructed with the sequences of the 3.0-kb fragments of SCBV showed that these SCBV isolates were dispersed in different phylogroups (Supplementary Fig. 1). Furthermore, two phylogenetic trees were constructed, one with the partial RT/RNase H sequences (about 800 bp) and another one with the promoter region sequences (644–934 bp) (Fig. 1). These trees included the sequences obtained in this study for eight virus isolates and the sequences of nine additional isolates retrieved from GenBank. The 17 SCBV sequences represented 14 SCBV genotypes (SCBV-A, SCBV-D to L, SCBV-N, SCBV-P, SCBV-Q, and SCBV-R) and were distributed in two major clades regardless of the sequence used to construct the trees. Clade I contained the eight isolates sequenced in this study and six isolates retrieved from GenBank (Fig. 1). Clade II was formed by three isolates retrieved from GenBank, namely SCBGAV-R570 (genotype A), SCBGAV-B51129 (genotype A), and SCBGDV-Batavia (genotype D). Overall, similar topology structure was observed among the three phylogenetic trees based on different fragment sequences. Based on the RT/RNase H sequences, nucleotide identity of clade I isolates varied from 74.0–93.0% and from 66.1–94.2% among isolates of clade II. Based on the promoter region sequences, nucleotide identity of clade I isolates varied from 38.8–93.9% and from 41.4–92.3% among isolates of clade II.

### Response to drought stress of seven SCBV promoters in transgenic *Arabidopsis* plants

To check if the promoters from different SCBV genotypes were active in monocots and dicots, eight SCBV promoters ($P_{SCBV-CHN1}$, $P_{SCBV-CHN2}$, $P_{SCBV-YZ2060}$, $P_{SCBV-FN39}$, $P_{SCBV-GT127}$, $P_{SCBV-FN2507}$, $P_{SCBV-TX}$ and $P_{SCBMOV-MOR}$) were tested in onion epidermal cells, young sugarcane leaves, and *Arabidopsis* leaf protoplasts. Transient *EYFP* expression was observed for all eight promoters and all tested plant cells or tissues (Supplementary Fig. 2). In T3 transgenic *Arabidopsis* lines, transformation events were obtained for seven of the eight promoters (no event obtained for $P_{SCBV-FN2507}$:GUS). Based on RT-qPCR data, the *GUS* expression level driven by $P_{SCBV-YZ2060}$ was similar to the one driven by the CaMV 35S promoter. This level associated with $P_{SCBV-YZ2060}$ was 94% higher than for promoter Ubi1 and >90% higher than for the six other SCBV promoters (Fig. 2a). Based on the fluorometric assay, high and similar GUS activity was obtained for $P_{SCBV-YZ2060}$:GUS, $P_{SCBV-CHN1}$:GUS, CaMV 35S:GUS, and Ubi1:GUS. GUS activity driven by $P_{SCBV-YZ2060}$ was >74% higher than the one driven by the five other promoters i.e. $P_{SCBV-CHN2}$, $P_{SCBV-FN39}$, $P_{SCBV-GT127}$, $P_{SCBV-TX}$, and $P_{SCBMOV-MOR}$ (Fig. 2b).

To identify SCBV promoters possessing drought-inducible characteristics, transgenic *Arabidopsis* lines were treated with 25% PEG6000. During the 3–24 h post treatment (hpt), transcript expression of the *GUS* gene was significantly upregulated by >360%, >380%, and >60% with SCBV promoters $P_{SCBV-YZ2060}$, $P_{SCBV-TX}$ and $P_{SCBV-CHN2}$, respectively (Fig. 3). No significant upregulation as compared to 0 hpt was found for the six other promoters ($P_{SCBV-GT127}$, $P_{SCBV-CHN1}$, $P_{SCBMOV-MOR}$, $P_{SCBV-FN39}$, CaMV 35 S, and Ubi1).

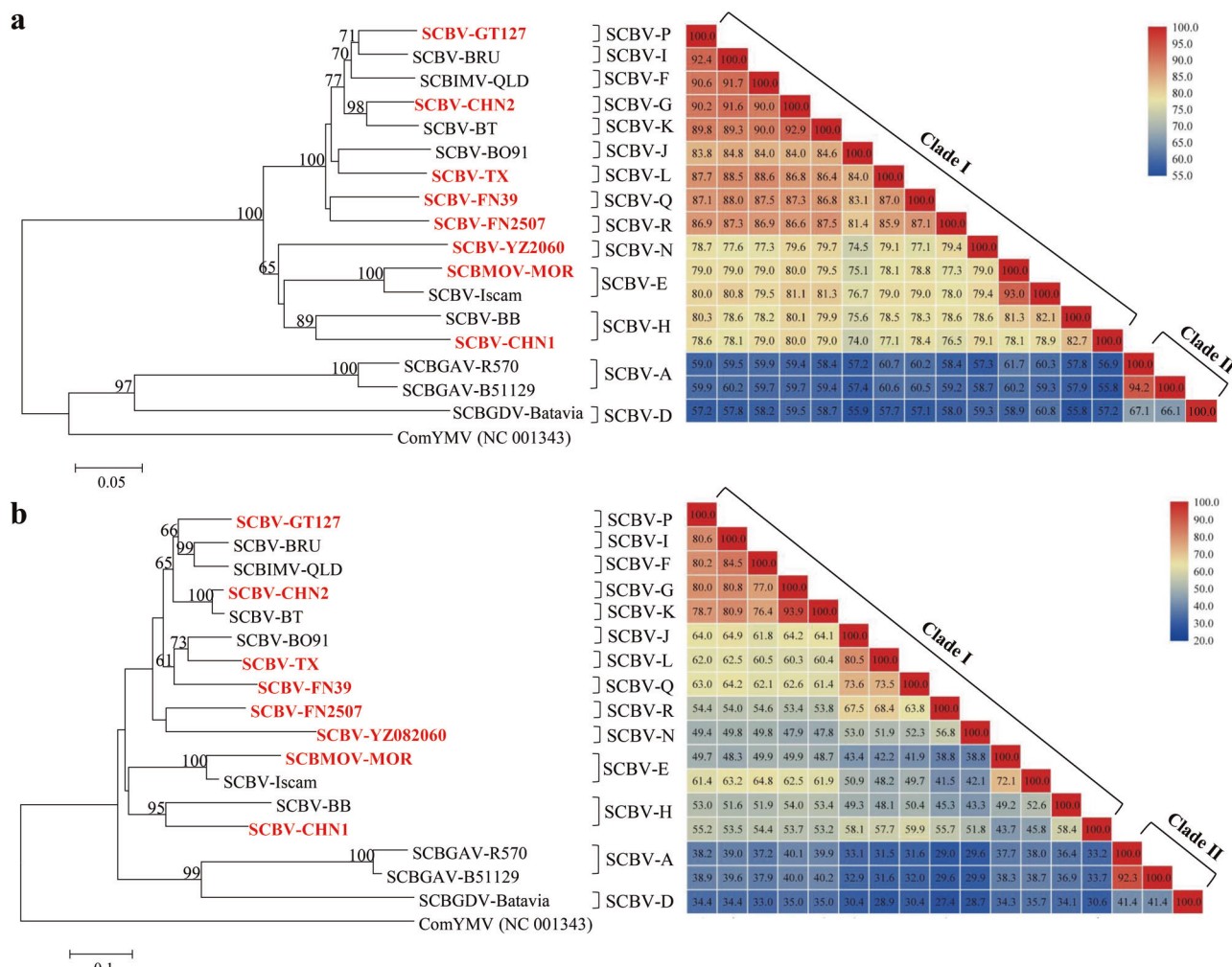

**Fig. 1 | Phylogenetic tree and pairwise identity matrix of SCBV isolates. a** Tree and matrix based on the partial sequence of the reverse transcriptase/ribonuclease H (RT/RNase H) genomic region (0.8 kb). **b** Tree and matrix based on the entire promoter region (0.6–0.9 kb) of 17 SCBV isolates. Isolates sequenced in this study are highlighted in red. The other nine isolates were retrieved from GenBank: SCBV-BRU (JN377537), SCBIMV-QLD (NC_003031), SCBV-BT (JN377536), SCBV-BO91 (JN377533), SCBV-Iscam (JN377534), SCBV-BB (JN377535), SCBGDV-Batavia (FJ439817), SCBGAV-R570 (FJ824813), SCBGAV-B51129 (FJ824814). Classification of SCBV genotypes was performed according to Janiga et al.[16]. Each phylogenetic tree was constructed using the neighbor-joining (NJ) method and bootstrap values (1000 replicates) are indicated at tree nodes. Bootstrap values below 60% were collapsed and ComYMV (NC 001343) was used as outgroup. Scale bar is in number of substitutions per nucleotide.

## Regulation of promoter $P_{SCBV-YZ2060}$ by ABA and drought treatments in transgenic plants and protoplasts of wild-type *Arabidopsis*

To investigate the response of promoter $P_{SCBV-YZ2060}$ to ABA and drought stress, transgenic *Arabidopsis* plants carrying $P_{SCBV-YZ2060}$:GUS were treated with 10 μM ABA or 25% PEG6000. GUS activity in these plants was significantly upregulated during 1–16 h after the treatment with 10 μM ABA (Fig. 4a). Highest GUS expression level was observed at 16 hpt, which corresponded to a 3.7-fold increase compared to the untreated control plants. GUS expression was also upregulated 3–24 h after the treatment of plants with 25% PEG6000 (Fig. 4b). Peak expression of GUS expression activity was at 12 hpt with a 3.3-fold increase compared to the untreated control plants.

Furthermore, GUS activity driven by promoters $P_{SCBV-YZ2060}$ from SCBV, $P_{ScRD29A}$ from sugarcane, and $P_{AtRD29A}$ from *Arabidopsis* was also significantly upregulated after an exogenous ABA treatment of wild-type *Arabidopsis* protoplasts. At 8 hpt, 2.7–6.4-fold increases were observed as compared to the non-treated controls (Supplementary Fig. 3a). Relative GUS activity driven by $P_{SCBV-YZ2060}$ was 2.2 and 5.5 higher than the one driven by promoters $P_{ScRD29A}$ from sugarcane and $P_{AtRD29A}$ from *Arabidopsis*, respectively (Supplementary Fig. 3b).

## Confirmation of the key domains and cis-acting elements in promoter $P_{SCBV-YZ2060}$ regulated by ABA and drought treatments

To identify the key domains and *cis*-acting elements in promoter $P_{SCBV-YZ2060}$ regulated by ABA and drought stress, an in silico analysis of the sequence of SCBV-YZ2060 was performed with the Neural Network Promoter Prediction program (NNPP, version 2.2). Two PPRs, namely PPR1 (74–124 nt) with transcription start site TSS1 and PPR2 (854–904 nt) with transcription start site TSS2 were identified (Supplementary Fig. 4). Furthermore, two TATA-boxes (AAATGA and ATAAGG) were found in PPR1 and PPR2, respectively. Two copies of ABA and drought-responsive ABRE motif (ABRE-2 and ABRE-1) were located at nt 735–740 and 801–807, respectively (Supplementary Fig. 4).

To map the active promoter region, a series of deletions were carried out around the two putative promoter regions (PPR1 and PPR2). The transient activity of the mutated promoter driving the *GUS* gene was tested in *Arabidopsis* protoplasts (Fig. 5a). When PPR1, PPR2, or both simultaneously were deleted in mutants $P_{YZ2060-S1}$, $P_{YZ2060-S2}$, $P_{YZ2060-S3}$, and $P_{YZ2060-S4}$, *GUS* expression was reduced by >90% compared to activity of the full-length promoter SCBV-YZ2060 (Fig. 5b). PPR1 and PPR2 promoter regions are therefore crucial domains of promoter $P_{SCBV-YZ2060}$.

**Fig. 2 | GUS expression triggered by different SCBV promoters in transgenic *Arabidopsis* plants. a** RT-qPCR assay of *GUS* gene, **b** Fluorometric GUS activity assay. Values are the means (±standard errors) of three T3 transgenic lines and three biological replicates. For each assay, mean values with the same letter are not significantly different at *P* = 0.05 according to Duncan's multiple range test.

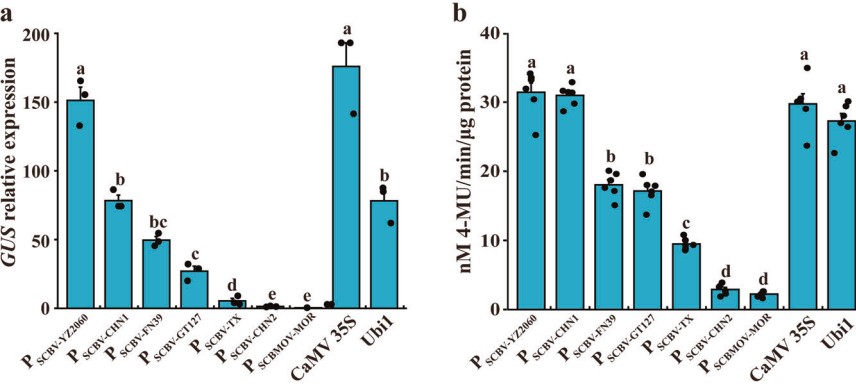

**Fig. 3 | Transcriptional expression of gene *GUS* driven by nine SCBV promoters in transgenic *Arabidopsis* plants subjected to a 25% PEG6000 treatment.** Values are the means (±standard errors) of three T3 transgenic lines and three biological replicates. For each promoter, mean values with the same letter are not significantly different at *P* = 0.05 according to Duncan's multiple range test.

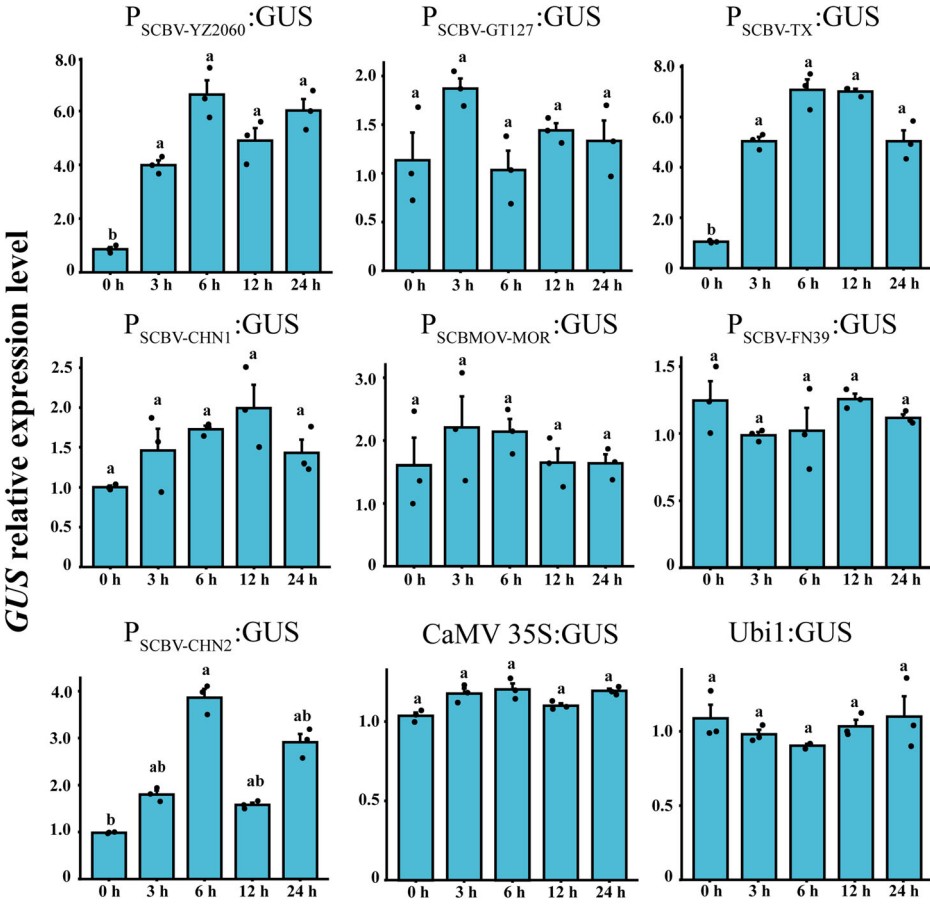

To determine the key domain of promoter $P_{SCBV-YZ2060}$ responding to ABA induction, the above-mentioned four mutated plasmids were also tested in *Arabidopsis* protoplasts subjected to 10 µM ABA treatment. Compared to untreated plants, GUS expression driven by mutant $P_{YZ2060-S4}$ was 2.9-fold higher after ABA treatment, while GUS expression driven by the other three mutants remained unchanged after ABA induction (Fig. 5c). Furthermore, to determine whether the two ABRE motifs are essential for the ABA-inducible elements, a series of deletions and point mutations were performed within ABRE-1 and ABRE-2. *GUS* transcript expression of mutants $P_{YZ2060-mL1}$:GUS, $P_{YZ2060-mL2}$:GUS, $P_{YZ2060-2mL}$:GUS, and $P_{YZ2060-2dL}$:GUS was unchanged in *Arabidopsis* protoplasts after ABA treatment. This suggested that the two ABREs of promoter SCBV-YZ2060 are the key cis-regulatory elements in response to ABA-induction (Fig. 5c).

To further investigate mutations of promoter SCBV-YZ2060 affected by ABA and drought stress treatments, transgenic *Arabidopsis* lines each

carrying one of eight mutant plasmids ($P_{YZ2060-S1}$:GUS, $P_{YZ2060-S2}$:GUS, $P_{YZ2060-S3}$:GUS, $P_{YZ2060-S4}$:GUS, $P_{YZ2060-mL1}$:GUS, $P_{YZ2060-mL2}$:GUS, $P_{YZ2060-2mL}$:GUS, and $P_{YZ2060-2dL}$:GUS) were produced using *Agrobacterium*-mediated transformation. *GUS* transcript expression of $P_{YZ2060-S4}$:GUS was upregulated 5.2 and 8.7 times in transgenic *Arabidopsis* after 25% PEG6000 and 10 µM ABA treatments, respectively, as compared to untreated control plants (Fig. 6a). Similar increases in upregulation were found in transgenic *Arabidopsis* transformed with wild promoter $P_{SCBV-YZ2060}$:GUS. No significant changes of GUS mRNA levels were observed in transgenic *Arabidopsis* plants carrying $P_{YZ2060-S1}$:GUS, $P_{YZ2060-S2}$:GUS, $P_{YZ2060-S3}$:GUS, $P_{YZ2060-mL1}$:GUS, $P_{YZ2060-mL2}$:GUS, $P_{YZ2060-2mL}$:GUS, and $P_{YZ2060-2dL}$:GUS, regardless of treatment (Fig. 6a). GUS protein activity of $P_{YZ2060-S4}$:GUS was 6.4-fold and 12.8-fold higher in transgenic *Arabidopsis* plants after 25% PEG6000 and 10 µM ABA treatments, respectively, as compared to untreated control plants (Fig. 6b).

**Fig. 4 | Expression of *GUS* gene driven by promoter P<sub>SCBV-YZ2060</sub> in transgenic *Arabidopsis* plants. a** Plants treated with 10 μM ABA. **b** Plants treated with 25% PEG6000. Values are the means (±standard errors) of three T3 transgenic lines and three biological replicates. For each treatment and each time point, values that are significantly different at $P = 0.05$ and $P = 0.01$ (Student's *T* test) are indicated by one and two asterisks, respectively.

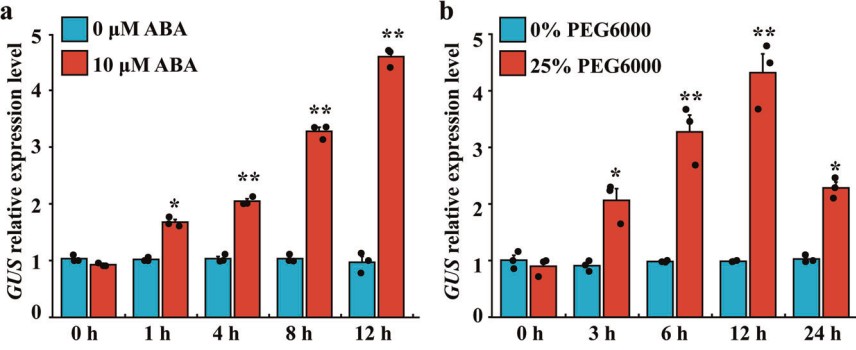

**Fig. 5 | Activity of promoter P<sub>SCBV-YZ2060</sub> and eight mutants of this promoter in *Arabidopsis* protoplasts. a** Schematic map of promoter P<sub>SCBV-YZ2060</sub> and eight of its mutants. T, *, and red dotted lines represent the transcription start site, the TATA box, and the deletion sites, respectively; + and – signs indicate presence and absence of *cis*-acting regulatory elements, respectively. Nucleotides written in lowercase letters represent the mutated nucleotides in the ABRE sequences. **b** Transcript expression of GUS gene driven by promoter P<sub>SCBV-YZ2060</sub> and four of its mutants. Blank control = no plasmid. **c** Transcript expression of *GUS* gene driven by promoter P<sub>SCBV-YZ2060</sub> and eight of its mutants subjected to 10 μM ABA treatment. Vector control = Ubi1:GUS. Values are the means (±standard errors) of four biological replicates and three technical replicates for each experiment. In **b**, mean values with the same letter are not significantly different at $P = 0.05$ according to Duncan's multiple range test. In **c**, and for each promoter, values that are significantly different at $P = 0.01$ (Student's *T* test) are indicated by two asterisks.

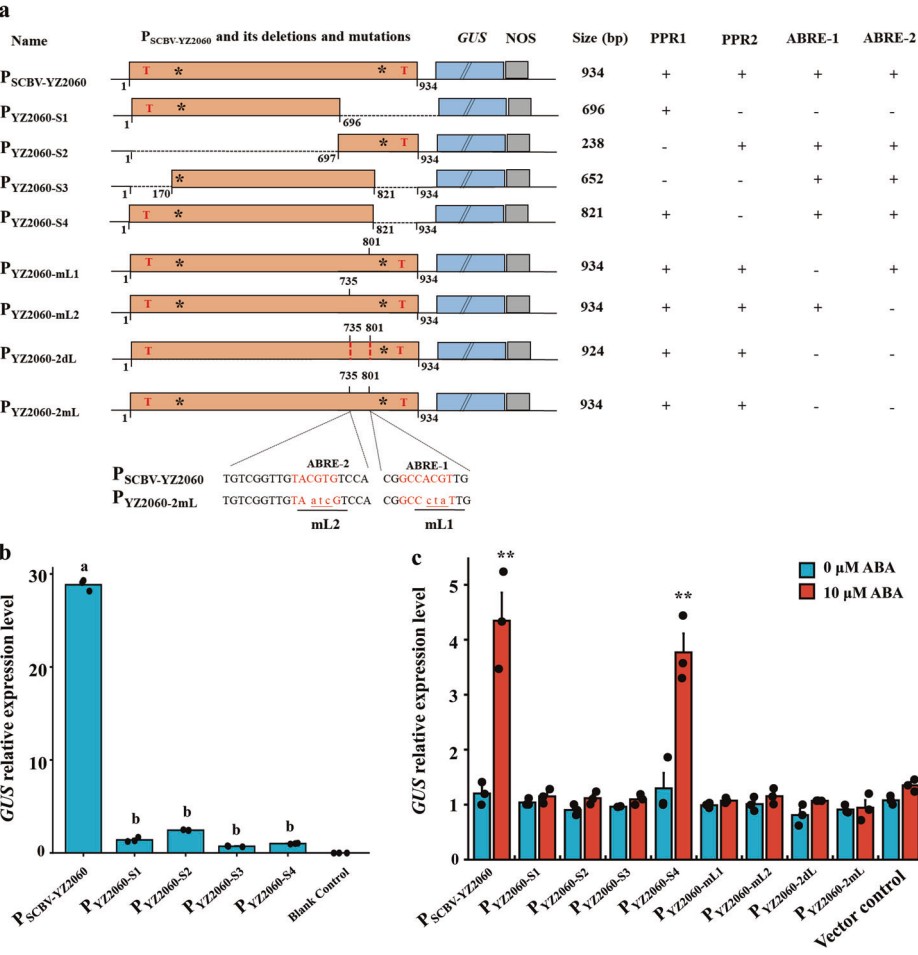

The overall GUS activity levels were, however, much lower for promoter mutant P<sub>YZ2060-S4</sub>:GUS than for P<sub>SCBV-YZ2060</sub>:GUS. No significant changes between untreated and treated plants were observed for the seven remaining promoter mutants (P<sub>YZ2060-S1</sub>:GUS, P<sub>YZ2060-S2</sub>:GUS, P<sub>YZ2060-S3</sub>:GUS, P<sub>YZ2060-mL1</sub>:GUS, P<sub>YZ2060-mL2</sub>:GUS, P<sub>YZ2060-2mL</sub>:GUS, and P<sub>YZ2060-2dL</sub>:GUS) (Fig. 6b).

### Regulation of transcriptional activity of promoter P<sub>SCBV-YZ2060</sub> from SCBV by transcription factor ScbZIP72 from sugarcane

To determine whether the TF bZIP protein recognizes and binds to an ABRE *cis*-acting element, a yeast one-hybrid assay was performed using two TFs, namely ScbZIP72 from sugarcane and AREB1 from *Arabidopsis*. A fragment named sL of 90 bp (located at nt 721–810) and another fragment named msL (a mutation of sL) containing ABRE-1 and ABRE-2 from

SCBV-YZ2060 were used as baits for the library screening (Fig. 7a). In the yeast one-hybrid assay, ScbZIP72 and AREB1 interacted with fragment sL possessing the wild-type ABRE motifs. No interaction was observed between ScbZIP72 or AREB1 and fragment msL with mutated ABRE motifs, thus indicating that ScbZIP72 from sugarcane and AREB1 from *Arabidopsis* can bind to ABRE motifs of promoter P<sub>SCBV-YZ2060</sub> in yeast cells (Fig. 7b). Additionally, the electrophoretic mobility shift assay (EMSA) indicated that TF ScbZIP72 binds in vitro to the ABRE motif in promoter P<sub>SCBV-YZ2060</sub> (Fig. 7c).

To further investigate the specific binding of plant TFs ScbZIP72 and AREB1 with viral promoter P<sub>SCBV-YZ2060</sub>, 4×sL (four tandem repetitions of AREB1) and 4×msL (mutated sequence of 4×sL) were fused upstream of P<sub>mini35S</sub>:GUS to generate reporter plasmids P<sub>4×sL-mini35S</sub>:GUS and P<sub>4×msL-mini35S</sub>:GUS, respectively. These constructions and the empty

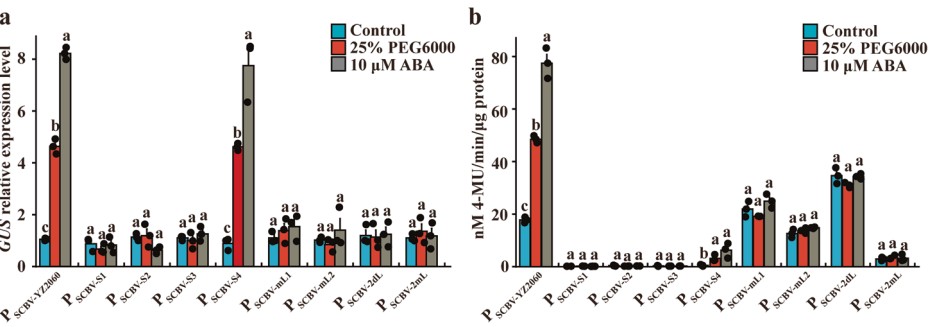

**Fig. 6 | Expression of *GUS* gene driven by promoter P_{SCBV-YZ2060} and eight of its mutants in transgenic *Arabidopsis* plants subjected to 25% PEG6000 and 10 μM ABA treatments. a** RT-qPCR assay of *GUS* gene, and **b** GUS protein activity assay. Control = no treatment. Values are the means (±standard errors) of three T3 transgenic lines and three biological replicates for each line. For each assay and for each promoter, mean values with the same letter are not significantly different at $P = 0.05$ according to Duncan's multiple range test.

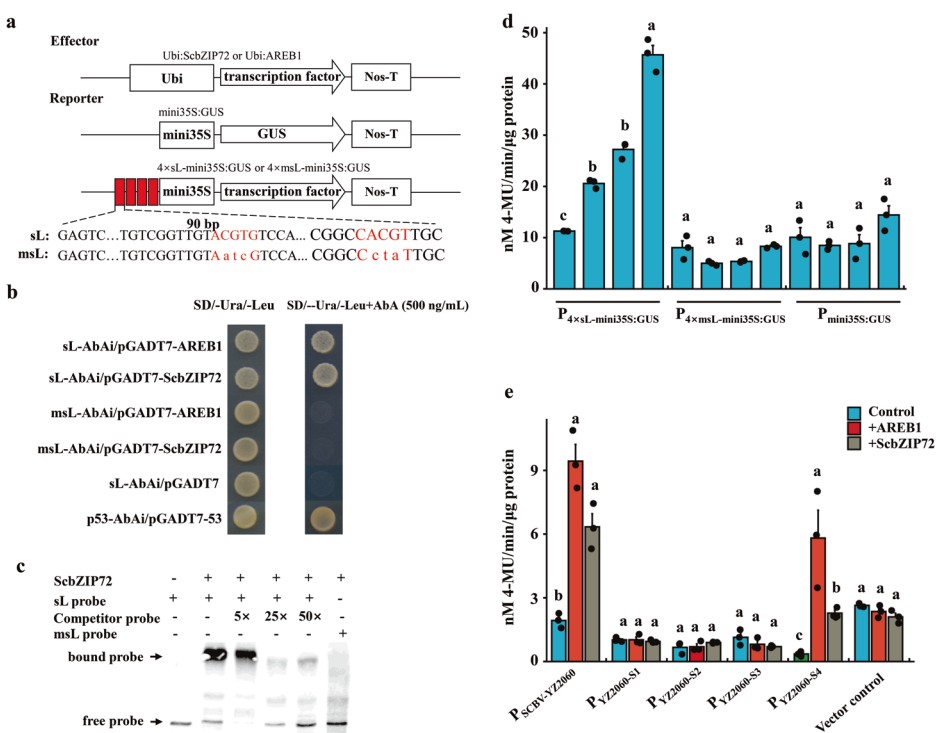

**Fig. 7 | Activation of viral promoter P_{SCBV-YZ2060} by transcription factors ScbZIP72 from sugarcane and AREB1 from *Arabidopsis*. a** Schematic map of the effector and reporter constructs: transcription factors ScbZIP72 from sugarcane or AREB1 from *Arabidopsis* in plasmid vectors Ubi:ScbZIP72 and Ubi:AREB1, P_{mini35S}:GUS vector, P_{4×sL-mini35S}:GUS = four tandem repetitions of AREB1 (sL) in P_{mini35S}:GUS vector, and P_{4×msL-mini35S}:GUS = msL (mutated sL) in P_{mini35S}:GUS. **b** Yeast one-hybrid assay with different combinations between ABRE and ScbZIP72 or AREB1: sL-AbAi/pGADT7-AREB1 = ABRE motif sL + AREB1; sL-AbAi/pGADT7-ScbZIP72 = ABRE motif sL + ScbZIP72; msL-AbAi/pGADT7-AREB1 = ABRE mutated motif msL + AREB1; msL-AbAi/pGADT7-ScbZIP72 = ABRE mutated motif msL + ScbZIP72; sL-AbAi/pGADT7 = negative control; p53-AbAi/pGADT7-p53 = positive control. **c** EMSA

showing that ScbZIP72 can directly target sL by binding to the ABRE motif. **d** GUS protein expression in *Arabidopsis* protoplasts of ABRE motif 4xsL of promoter P_{SCBV-YZ2060} and mutants of this promoter motif after 10 μM ABA treatment (6 h) or co-transformation with bZIP transcription factors AREB1 and ScbZIP72. Control = no treatment. **e** GUS protein expression in *Arabidopsis* protoplasts of promoter P_{SCBV-YZ2060} and mutants of this promoter after co-transformation with bZIP plant transcription factors (AREB1 or ScbZIP72). Vector control = Ubi1:GUS. Values are the means (±standard errors) of four biological replicates and three technical replicates for each experiment. For each assay and for each vector or promoter, mean values with the same letter are not significantly different at $P = 0.05$ according to Duncan's multiple range test.

vector P_{mini35S}:GUS were used as reporters whereas Ubi:ScbZIP72 and Ubi:AREB1 were used as effectors. Transient GUS expression assays in wild-type *Arabidopsis* protoplasts were performed after transformation with the GUS reporters (and subjected to ABA treatment) or co-transformation with the GUS reporters and the effector plasmids Ubi:ScbZIP72 and Ubi:AREB1 (Fig. 7d). GUS activity of P_{4×sL-mini35S}:GUS was significantly increased (81–303%) in *Arabidopsis* protoplasts after 10 μM ABA treatment or co-transformation with effectors ScbZIP72 and AREB1, as compared to untreated control transformed with P_{4×sL-mini35S}:GUS only. No significant changes of GUS activity were observed for reporters P_{4×msL-mini35S}:GUS or P_{mini35S}:GUS subjected to the same ABA treatment or co-transformation of *Arabidopsis* protoplasts (Fig. 7d).

To determine whether ScbZIP72 and AREB1 can transactivate promoter P_{SCBV-YZ2060}, effectors Ubi:ScbZIP72 and Ubi:AREB1 were each co-transfected into *Arabidopsis* protoplasts with P_{SCBV-YZ2060}:GUS and four mutants of this promoter vector, namely P_{YZ2060-S1}:GUS, P_{YZ2060-S2}:GUS, P_{YZ2060-S3}:GUS, and P_{YZ2060-S4}:GUS (Fig. 7e). When compared to the control (no effector), relative GUS activity for promoters P_{SCBV-YZ2060} and P_{SCBV-YZ2060-S4} was increased 3.8 and 16.5 times, respectively, in the presence of effector Ubi:AREB1. Similarly, relative GUS activity of the two promoters was upregulated 2.3 and 5.4 times in the presence of effector Ubi:ScbZIP72 (Fig. 7e). Relative GUS activities measured for constructions P_{YZ2060-S1}:GUS, P_{YZ2060-S2}:GUS, and P_{YZ2060-S3}:GUS were not significantly changed in presence ScbZIP72 or AREB1, as compared to the respective controls without effectors (Fig. 7e). This suggested that TFs AREB1 and

ScbZIP72 activate and enhance activity of promoter $P_{SCBV-YZ2060}$ only when PPR1 and ABRE motifs were both present.

## Drought-inducible activity conferred by promoter $P_{SCBV-YZ2060}$ in transgenic sugarcane

To investigate the expression pattern of promoter $P_{SCBV-YZ2060}$ in sugarcane, 55 transgenic sugarcane lines were generated by transforming cultivar ROC22 with construct $P_{SCBV-YZ2060}$:GUS. Twenty-one positive plants were verified by four detection methods, namely PCR targeting the SCBV promoter and the *GUS* gene, PAT/bar strip test, RT-qPCR for *GUS* transcription expression, and GUS fluorometric assay (Supplementary Fig. 5). When transformed sugarcane plants had 5–6 unfolded leaves, GUS activity was determined in root, stem, and leaf tissue of four transgenic lines (ScL5, ScL14, ScL33 and ScL44). Highest (2.49 nM 4-MU/min/µg protein) and lowest (0.46 nM 4-MU/min/µg protein) GUS activity was found in root and leaf tissue, respectively (Supplementary Fig. 6). This suggested that promoter $P_{SCBV-YZ2060}$ promoter could have tissue-specific patterns.

To analyze the response of promoter $P_{SCBV-YZ2060}$ to drought stress and ABA in sugarcane, six lines transformed with $P_{SCBV-YZ2060}$:GUS were selected and treated with 25% PEG6000 or 10 µM ABA. *GUS* expression determined by RT-qPCR was significantly upregulated after the PEG6000 and ABA treatments, but GUS mRNA levels varied according to leaf tissue (Fig. 8a). *GUS* gene expression in roots was 17.8 and 9.6 times higher in plants subjected to drought stress and ABA treatment, respectively, than in untreated control plants. In leaf tissue, this increase was by factors 4.4 and 8.3, respectively. No increase of *GUS* expression was detected in stem tissue after PEG6000 treatment, but the GUS mRNA level increased 20.7 times in stems of ABA treated plants. GUS protein activity was also significantly increased in root and leaf tissues after PEG6000 and ABA treatments (Fig. 8b). It was increased in stem tissue after application of PEG6000 but not after ABA treatment.

Finally, transcriptional expression levels of gene *ScbZIP72* were determined by RT-qPCR in six sugarcane transgenic lines (Supplementary Fig. 7). After 25% PEG6000 treatment, *ScbZIP72* mRNA levels were significantly increased 4.7 and 4.1 times in root and leaf tissues, respectively. After ABA treatment, these expression levels were significantly increased 20.2 and 26.3 times in root and leaf tissues, respectively. No significant changes in transcription levels of gene *ScbZIP72* were observed in stem tissue, regardless of treatment.

## Discussion

Expression of a foreign gene and accumulation of the transgenic protein only under specific environmental conditions and/or in specific tissues are the desired goals of plant engineering. The transgenic approach is an effective strategy for generating drought-resistant plants using drought-inducible promoter(s) driving resistant gene expression[6]. In petunia, over-expression of gene *LeNCED1* (related to the ABA biosynthesis pathway) with inducible promoter *rd29A* improved drought resistance and lacked negative pleiotropic effects on plant growth and development that was observed with constitutive promoter CaMV *35S*[42]. Because endogenous viral

elements are widespread in plant genomes and because badnaviruses can be integrated in their host genome, SCBVs that exhibit high genetic diversity are good candidates for identification of alternative promoters and their use in plant genetic engineering[9,16,43]. The features of stress-expression of SCBV promoters remain unclear.

In this study, genome fragments from eight different SCBV genotypes were amplified, cloned, and sequenced. These fragments contained RT/RNase H (~800 nt) and a promoter region (~900 nt). RT/RNase H sequences had 77–90% nucleotide identity and the sequences of the promoter region had only 48–80% nucleotide identity. These data support the high genetic diversity of the promoter region among SCBV isolates[9], which provides a choice to investigate desired promoters from this virus for driving targeted genes in transgenic sugarcane, an allopolyploid crop. *GUS* expression driven by various SCBV promoters differed in transgenic plants. For example, the promoter (a 1.4 kb DNA fragment) from isolate SCBMOV-MOR was responsible for constitutive gene expression in transgenic tobacco and *Arabidopsis*[18,19], while it showed tissue-specificity in banana, oat, barley, and wheat[20]. The promoter of isolate SCBIMV-QLD was predominantly expressed in the meristematic regions of the stem[21], while promoter SCBV21 from isolate SCBV-TX was specifically expressed in culm vascular bundles in transgenic sugarcane[9].

In this study, trimmed promoter $P_{SCBMOV-MOR}$ that was harbored on a 0.6 kb DNA fragment exhibited very low GUS activity in comparison to previous studies using larger DNA fragments (1.4 kb DNA fragment)[18]. This result was most likely due to the absence of a TATA-box and multiple promoter enhancer elements (CAAT-box) in the smaller fragment. The core promoter sequence consists of a transcription start site and a TATA-box, with various *cis*-regulatory elements interacting with TFs[7,44]. The PPR2 region is critical for promoter activity of SCBMOV-MOR[18,20], SCBIMV-QLD[22] and SCBV21[9]. GUS activity significantly increased when enhancer sequences (without PPR2 region) from the promoter of SCBIMV-QLD were fused with the truncated maize alcohol dehydrogenase 1 (*ZmADH1*) promoter[22]. Notably, promoter $P_{SCBV-YZ2060}$ contained two PPR regions (PPR1 and PPR2) that were closely linked to promoter activity.

In plants, tolerance to drought is regulated by multiple genes and several endogenous hormones such as ABA[26,45]. In this study, three SCBV promoters ($P_{SCBV-YZ2060}$, $P_{SCBV-TX}$, and $P_{SCBV-CHN2}$) were demonstrated for the first time to be inducible by drought stress in transgenic *Arabidopsis*. PPR1 and two ABRE (ABRE-1 and ABRE-2) were identified as key domains of promoter $P_{SCBV-YZ2060}$ responding to ABA induction and drought stress. ABRE contains a core sequence (ACGT) that is recognized by bZIP family members, and these TFs are strongly induced by ABA and drought stress[26,46]. ABREs are major *cis*-acting elements in ABA-dependent signaling pathways regulating expression of ABA-responsive genes in plants subjected to osmotic stress[47]. Promoter $P_{SCBV-YZ2060}$ contains two ABA response elements (ABRE-1 and ABRE-2) which were indispensable for this promoter to respond to drought stress and ABA induction. ABREs responding to ABA were reported previously in other viruses and in plants. For example, the ABREs occur in promoters of horseradish latent virus (HRLV)[48] and mungbean yellow mosaic virus (MYMV)[49], and in promoters

**Fig. 8 | Expression of *GUS* gene in root, stem, and leaf tissue of sugarcane transformed with promoter construct $P_{SCBV-YZ2060}$:GUS and subjected to 25% PEG6000 and 10 µM ABA treatments.**
**a** RT-qPCR of *GUS* gene and **b** GUS protein activity assay. Control = no treatment. Values are the means (±standard errors) of six transgenic lines and three technical replicates for each line. For each assay and for each sugarcane tissue, mean values with the same letter are not significantly different at *P* = 0.05 according to Duncan's multiple range test.

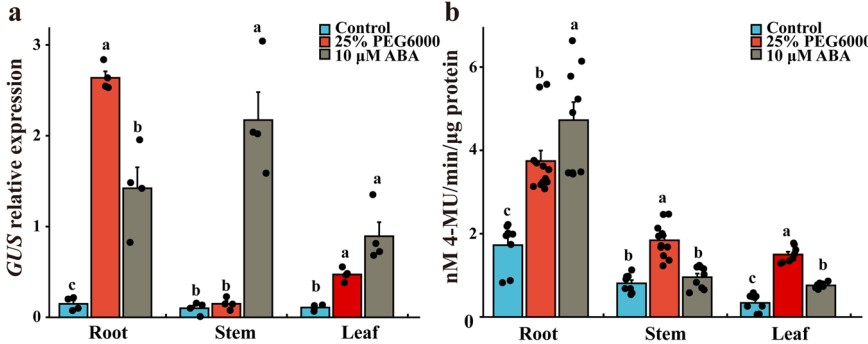

of the pyrroline-5-carboxylate synthetase (P5CS) gene from barley[50] and *Arabidopsis*[51]. These ABREs-including promoters were also responsive to ABA treatment and/or drought stress[48,50,51].

Furthermore, Y1H and EMSA assays revealed that TFs ScbZIP72 from sugarcane and AREB1 from *Arabidopsis* did bind to ABRE of promoter $P_{SCBV-YZ2060}$. Both TFs activated transcription of $P_{SCBV-YZ2060}$ promoter in *Arabidopsis* protoplasts. Transcriptional regulation of gene expression under osmotic stress conditions such as drought and high salinity is governed by two key *cis*-acting elements, ABREs and MYB (myeloblastosis). These two elements are involved in the ABA-dependent signal pathway and the C-repeat (CRT)/dehydration-responsive element (DRE) participates in the ABA-independent signal pathway[27]. ABA-dependent kinase proteins were previously proposed to activate or stabilize bZIPs proteins by phosphorylation and bind to *cis*-acting elements (e.g., ABRE and/or DRE), thus controlling gene expression in sugarcane[52] and other plants[53]. Additionally, modulation of the bZIP proteins in ABA signaling and drought responses occurs by ubiquitination and sumoylation[54]. Transcription factor OsbZIP20 interacts directly and is phosphorylated by the SAPK10 protein (a member of SnRK2s family) before binding to the ABRE element of the promoter of NHX1 [Na(+)(K(+))/H(+) exchanger 1][30]. Consequently, NHX1 transcription is regulated, and this contributes to enhancement of rice tolerance to drought and salt.

AREB1 belongs to the bZIP group-A subfamily and is activated in response to abiotic stress and ABA treatments[55,56]. In *Arabidopsis*, AREB1 specifically binds to ABRE of promoter *RD29B*, thus regulating the expression of *RD29B* and enhancing drought resistance of transgenic plants[54]. In *Populus trichocarpa*, PtrAREB1-2 binds to ABRE motifs in promoters of three drought-responsive *PtrNAC* genes and influences histone acetylation of ABRE motifs, resulting in activation of these *PtrNAC* genes and regulation of drought response and tolerance[56]. In our study, $P_{SCBV-YZ2060}$ acted like an ABA-inducted and drought stress promoter in monocot (sugarcane) and in dicot (*Arabidopsis*) plants with sugarcane ScbZIP72 and *Arabidopsis* AREB1 binding to ABREs of promoter $P_{SCBV-YZ2060}$, thus suggesting that this mechanism was conserved between the two species (Fig. 9). Molecular interactions of ScbZIP72 and AREB1 with ABRE motifs in the SCBV promoter fused with the resistant gene for enhancement of plant tolerance to abiotic stresses remain to be investigated.

To gain robust agriculture products using less land and to meet the needs of increasing world populations, engineering plants by stacking multiple genes in transgenic crops might become essential in the future[11,57]. However, homology-dependent gene silencing can disrupt transgene expression in transgenic plants carrying multiple transgenes driven by the same promoters[7,37]. Use of naturally occurring allelic promoters in a transgenic crop such as sugarcane, that possesses a highly polyploid and heterozygous genome, may also result in inhibition of transgene expression because of accrued defects in promoters and transcriptional gene silencing[10]. Transgene silencing in sugarcane is not simply triggered by multiple copies of the promoter and 5' untranslated leader sequences[40]. New approaches to avoid transgene silencing are essential to obtain the desired patterns of expression in a high number of transgenic lines. In addition to exploring more alternative promoters from plant viruses, optimizing native promoter elements as well as generating synthetic ones with desirable features in plant engineering is also promising[7,57]. Interestingly, SCBV is a virus naturally hosted by sugarcane and it infects a limited number of monocot plants including banana, rice, *Sorghum halepense* and *Brachiaria extensa*[43]. Nevertheless, promoters from SCBVs have broad application prospects in plant transgenics, including monocot and dicot plants[9,19,21]. Based on our investigations, $P_{SCBV-YZ2060}$ and its *cis*-acting elements could be a valuable genetic tool for regulated transgene expression during drought stress.

In summary, activity of seven promoters from different SCBV genotypes varied in transgenic *Arabidopsis* plants. Three of these promoters, namely $P_{SCBV-YZ2060}$, $P_{SCBV-TX}$, and $P_{SCBV-CHN2}$ were proven for the first time to be drought-induced promoters. GUS activity driven by promoter $P_{SCBV-YZ2060}$ was also significantly induced by ABA. PPR1 and two ABREs (ABRE-1 and ABRE-2) were essential for regulation of drought stress and ABA induction by this promoter. ABRE-1 and ABRE-2 of promoter $P_{SCBV-YZ2060}$ were able to bind with two bZIP subfamily, AREB1 from *Arabidopsis* and newly identified ScbZIP72 from sugarcane. In transient expression assays, $P_{SCBV-YZ2060}$ was activated after co-infection of *Arabidopsis* protoplasts with either ScbZIP72 or AREB1. GUS expression was also significantly upregulated in sugarcane plants transformed with $P_{SCBV-YZ2060}$:GUS and submitted to ABA treatment or drought stress. Additionally, GUS activity driven by $P_{SCBV-YZ2060}$ was more strongly upregulated by an ABA treatment when compared to the native promoters $P_{ScRD29A}$ from sugarcane and $P_{AtRD29A}$ from *Arabidopsis*. These results suggested that this viral promoter might be more suitable for driving strong expression of a transgene in response to drought stress than the drought-responsive promoters from sugarcane and *Arabidopsis*. These findings could lead to an alternative strategy to generate transgenic sugarcane with drought-tolerance based on promoter $P_{SCBV-YZ2060}$ or $P_{YZ2060-S4}$ driving TF *ScZIP72* or other drought-resistance genes. Additionally, the core *cis*-acting elements of $P_{SCBV-YZ2060}$ identified in this study could be used for artificial synthetic promoter designs to engineer

**Fig. 9 | Model for regulation in transgenic plants of promoter $P_{SCBV-YZ2060}$ by transcription factors AREB1 from *Arabidopsis* and ScbZIP72 from sugarcane modulating drought and ABA stress responses.** Red arrows represent positive regulation. Red triangles indicate the locations of ABRE cis-acting elements in promoter $P_{SCBV-YZ2060}$.

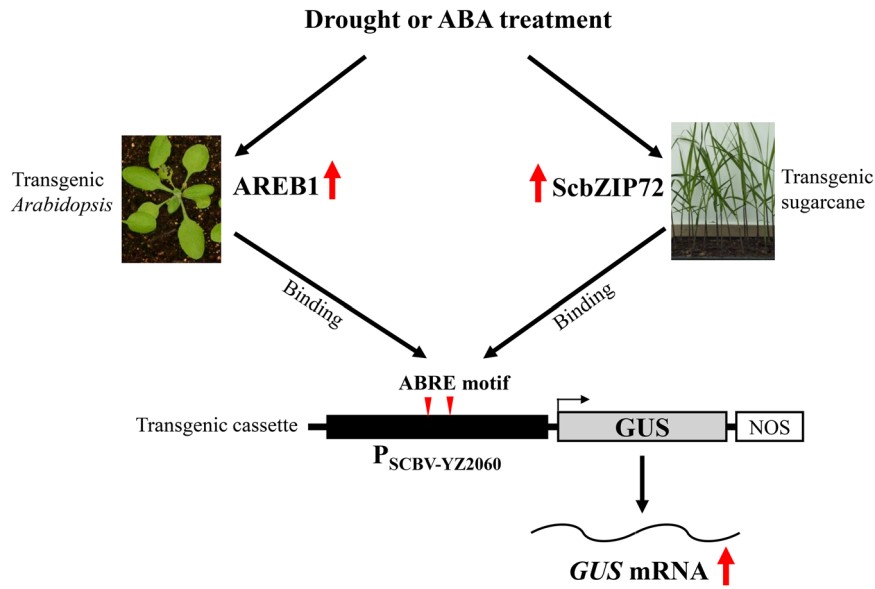

plants for production of robust agricultural outputs tolerating drought stress.

## Materials and methods

### Isolation and sequence analysis of SCBV promoters

Genomic fragments (about 3000 bp each) of SCBV were amplified from six sugarcane cultivars infected by different virus isolates, namely SCBV-CHN2 (genotype G), SCBV-CHN1 (genotype H), SCBV-YZ2060 (genotype N), SCBV-FN39 (genotype Q), SCBV-GT127 (genotype P), and SCBV-FN2507 (genotype R) from sugarcane cultivars CZ66-70, ROC27, YZ08-2060, FN39, GT88-127, and FN02-2507, respectively. The PCR assay was performed with degenerated primers SCBV-AF5603 and SCBV-AR1002 (Supplementary Table 1) in a total reaction volume of 50 μL using *LA* Taq polymerase (TaKaRa, Dalian, China), following the protocol developed previously[14]. The six genomic fragments of SCBV were cloned into vector pMD19-T, and then verified by sequencing[14]. Two other SCBV promoter region sequences (about 1000 bp each) were also subcloned into vector pMD19-T from plasmid pScBV20 (virus isolate SCBMOV-MOR, genotype E)[58] and plasmid pSCBV21:GUS (virus isolate SCBV-TX, genotype L)[9]. All the constructs were verified by Illumina DNA sequencing. The nucleotide sequences of the six amplified SCBV fragments were deposited at NCBI GenBank under accession numbers KM214357-KM214358 and OL413029-OL413032.

The PPRs and transcription start sites (TSS) of SCBV promoters were identified in silico with the Neural Network Promoter Prediction program (http://www.fruitfly.org/seq_tools/promoter.html)[59]. Putative *cis*-acting elements were predicted by PlantCARE (http://bioinformatics. psb.ugent.be/webtools/plantcare)[60] and plant *cis*-acting regulatory DNA elements were determined using the PLACE database (https://www.dna. affrc.go.jp/PLACE/?action=newplace)[61]. Nucleotide sequences of the reverse transcriptase/ribonuclease H (RT/RNase H) and the promoter region of the eight virus isolates mentioned above and nine additional isolates retrieved from GenBank (Fig. 1) were aligned with the ClustalW algorithm implemented in MEGA 7.0[62]. Nucleotide sequence identities were estimated by pair-wise sequence comparison using the BioEdit program[63]. The pairwise identity figure of sequences was drawn with the sequence demarcation tool (SDT) software using MUSCLE alignment. The neighbor-joining (NJ) phylogenetic trees were constructed with the MEGA7.0 program based on RT/RNase H and the promoter regions sequences. Corresponding sequences from an isolate of commelina yellow mottle virus (ComYMV, GenBank no. NC001343) were used as outgroups. The robustness of the trees was determined by bootstrap analysis (1000 replicates) and the bootstrap values were indicated at the tree nodes.

### Plasmid construction of the SCBV and plant native promoters and their mutants

The approximately 0.9 kb sequences of the SCBV promoter was amplified from the eight SCBV genomic fragments by PCR using sequence-specific primers (Supplementary Data 1). The PCR program consisted of one cycle at 98 °C for 2 min, followed by 35 cycles of 10 s denaturation at 98 °C, 15 s annealing at 60 °C, and 2 min extension at 72 °C; and a final cycle at 72 °C for 10 min. The CaMV 35 S promoter of the backbone of vector CaMV 35S:EYFP-NOS/pSK (digested with *Xho*I and *Nco*I) was replaced by each SCBV promoter using the In-Fusion HD Cloning Kit (TaKaRa). This resulted in production of eight plasmids named $P_{SCBV-CHN2}$:EYFP, $P_{SCBV-CHN1}$:EYFP, $P_{SCBV-YZ2060}$:EYFP, $P_{SCBV-FN39}$:EYFP, $P_{SCBV-GT127}$:EYFP, $P_{SCBV-FN2507}$:EYFP, $P_{SCBV-TX}$:EYFP, and $P_{SCBMOV-MOR}$:EYFP. The strong constitutive promoters of CaMV 35S:EYFP and Ubi1:EYFP were used as controls[9]. The same procedure was used to replace Ubi1 in the Ubi1:GUS vector (digested with *Hin*dIII and *Bam*HI) with the eight SCBV promoters, resulting in production of plasmids $P_{SCBV-CHN2}$:GUS, $P_{SCBV-CHN1}$:GUS, $P_{SCBV-YZ2060}$:GUS, $P_{SCBV-FN39}$:GUS, $P_{SCBV-GT127}$:GUS, $P_{SCBV-FN2507}$:GUS, $P_{SCBV-TX}$:GUS, and $P_{SCBMOV-MOR}$:GUS. Ubi1:GUS and CaMV 35S:GUS plasmids were used as controls[9].

Deletions of the SCBV-YZ2060 promoter and mutations of the ABRE motif of this promoter were generated in plasmid $P_{SCBV-YZ2060}$:GUS using the overlap PCR extension. Deletion fragments of Δnt697–nt934 (YZ2060-S1), Δnt1–nt697 (YZ2060-S2), Δnt1–nt179 (YZ2060-S3), and Δnt822–nt934 (YZ2060-S4) were amplified and individually cloned into the backbone of vector Ubi1:GUS digested with *Hin*dIII and *Bam*HI using the In-Fusion HD Cloning Kit. This resulted in production of four recombinant constructs named $P_{YZ2060-S1}$:GUS, $P_{YZ2060-S2}$:GUS, $P_{YZ2060-S3}$:GUS, and $P_{YZ2060-S4}$:GUS. Single ABRE motif mutation ABRE-1 (mL1) and ABRE-2 (mL2), double ABRE motif mutation (mL) and double ABRE motif deletion (dL) were individually cloned into the above-mentioned backbone of vector Ubi1:GUS to generate recombinant constructs $P_{YZ2060-mL1}$:GUS, $P_{YZ2060-mL2}$:GUS, $P_{YZ2060-dL}$:GUS, and $P_{YZ2060-mL}$:GUS. Promoter SCBV-YZ2060 was also used to insert the fragment of four tandem repetitions of the ABRE motif (4×sL) and the fragment of four tandem repetitions of a signal mutation of the ABRE motif (4×msL) in front of the minimum 35S promoter (mini35S, 100 bp core region)[64]. To reach this goal, a modified *GUS* expression vector ($P_{mini35S}$:GUS) was produced from CaMV 35S:GUS and used to generate constructs $P_{4×sL-mini35S}$:GUS and $P_{4×msL-mini35S}$:GUS after *Hin*dIII digestion. The expected sequences of all constructs was confirmed by Illumina DNA sequencing.

Additionally, the full-length ORFs of ScbZIP72 and AREB1 were cloned from sugarcane cultivar ROC22 and *Arabidopsis* ecotype Columbia-0, respectively. Each ORF was separately inserted into empty vector Ubi-nos[36] digested with *Sal*I to obtain plasmids Ubi:ScbZIP72 and Ubi:AREB1. Furthermore, two promoter sequences of $P_{ScRD29A}$ (about 2.0 kb) and $P_{AtRD29A}$ (about 0.45 kb) were amplified from the above-mentioned plants (sugarcane and *Arabidopsis*) by PCR with two sequence-specific primers (Supplementary Table 1). Subsequently, the two trimmed promoters of $P_{ScRD29A}$ (729 bp) and $P_{AtRD29A}$ (458 bp) replaced the Ubi1 in the backbone of plasmid Ubi1:GUS and then resulted into $P_{ScRD29A}$:GUS and $P_{AtRD29A}$:GUS plasmids, respectively. The sequences of all plant-expression vectors constructed in this study are registered in the NCBI library under accession numbers OL322080-OL322089.

### Transient expression in plant cells and tissues

The constructs of reporter gene *EYFP* with different promoters were introduced into onion epidermal cells and young sugarcane leaf fragments by microprojectile bombardment (PDS-1000/He; Bio-Rad Laboratory, California, USA)[9]. After incubation at 28 °C for 48 h, EYFP fluorescence in the targeted tissues was observed using a stereomicroscope (SteREO Lumar.V12, Zeiss, Oberkochen, Germany) with 25× and 40× amplifications and YFP filters. The protoplasts derived from *Arabidopsis* mesophyll cells were isolated and used for the transient expression assay following the protocol described previously[64,65]. After transfection of plasmids into *Arabidopsis* protoplasts using the PEG-CaCl₂ method, protoplasts were cultured for 2 h at room temperature prior addition of 10 μM ABA to each sample. These protoplasts were maintained for 8 h at room temperature before performing the GUS activity assay. The experiments were performed using three independent replicates.

### Generation of transgenic plants and drought-stress treatments

Transgenic plants of *Arabidopsis* ecotype Columbia-0 were generated by the floral dip method using *Agrobacterium tumefaciens*[66]. The recombinant plasmids were introduced into *A. tumefaciens* strain GV3101 following the manufacturer's protocol (Weidi Bio, Shanghai, China). These constructs contained the *PAT/bar* gene cassette that originated from the backbone of CaMV 35S:GUS to enable Basta (glufosinate ammonium)-based plant selection. The surviving transformants (T1) were confirmed by PCR detection of the SCBV promoter and RT-PCR amplification of *GUS* mRNA. Seeds from three T1 plants were used to multiply each transformant for two generations. The seeds of T3 transgenic *Arabidopsis* were collected individually and used for further experiments as follows. Three T3 transgenic lines were analyzed for each experiment. *Arabidopsis* transgenic plants were grown in pots containing a mixture of organic substrate and vermiculite

(3/1, v/v) or in Petri dishes containing Murashige and Skoog (MS) medium supplemented with 1% sucrose. Plants were maintained in a growth chamber at 22 °C with a photoperiod of 16 h/8 h light/dark. *Arabidopsis* seeds were surface-disinfected and then vernalized by keeping them at 4 °C for two days. Roots of two-week-old *Arabidopsis* seedlings were dipped into 25% PEG6000 for 0–24 h or 10 μM ABA for 0–16 h. After application of dehydration treatment with PEG6000, whole plantlets were sampled at 0 h, 3 h, 6 h, 12 h, and 24 h. The entire plants were sampled at 0 h, 1 h, 4 h, 8 h, and 16 h after the treatment with ABA. All samples were immediately frozen in liquid nitrogen and stored at −80 °C until further use.

Transgenic sugarcane carrying the $P_{SCBV-YZ2060}$:GUS cassette was generated through microprojectile bombardment (PDS-1000/He, Bio-Rad, USA) following the protocol described previously[9]. The embryogenic calli generated from sugarcane cultivar ROC22 were used as explants. The surviving plants were collected for further analysis after serial Basta-treatment selection. Transgenic plants were grown in a growth chamber at 28 °C with 16 h light and 8 h dark until production of 2–3 fully expanded leaves. Roots of plants were dipped into 25% PEG6000 or 10 μM ABA solutions. Leaf, stem, and root tissues were collected 12 h after application of the dehydration (PEG6000) or ABA treatment. All samples were immediately frozen in liquid nitrogen and stored at −80 °C until further use.

### PAT/bar strip test
To test the bar protein expression of putative transgenic lines of sugarcane, the fresh leaves (0.1 g) of the putative transgenic and wild-type plantlets were collected and homogenized in a 1.5 mL microcentrifuge tube. The PAT/bar strip (QuickStix for PAT/bar, Envirologix, Portland, USA) was dipped into tissue homogenates for 5 min following the manufacturer's protocol.

### Histochemical and fluorometric GUS assays
Histochemical analysis of GUS activity was performed based on the protocol described previously[67]. The samples were vacuum-filtered for 15 min and then incubated overnight in the GUS reaction mixture at 37 °C in the dark. The samples were then incubated in 70% ethanol for 12 h to remove chlorophylls and pigments. Images were taken with a dissecting microscope (Leica EZ4, Leica, Germany). The fluorometric GUS assay was performed as previously described[9]. Fluorometric GUS was quantified using a Synergy™ H1 Hybrid Multi-Mode Reader (BioTek, Vermont, USA) with emission at 455 nm and excitation at 365 nm. Total proteins were extracted with the Plant Protein Extraction Kit (Solarbio, Beijing, China) according to the manufacturer's protocol. The total protein concentration was determined with the BCA Protein Assay Kit (Solarbio). Three transgenic lines and three individual plants (biological replicates) per transgenic line were analyzed for each treatment.

### RNA extraction and quantitative real-time PCR analysis
Total RNA was extracted from *Arabidopsis* and sugarcane tissue using the TRIzol reagent (Invitrogen, Carlsbad, USA) according to the manufacture's protocol. The quality of each RNA sample was determined by electrophoresis using 1% agarose gels and RNA concentration was measured with a Synergy™ H1 Hybrid Multi-Mode Reader (BioTek). All RNA samples were treated with Dnase I (TaKaRa) and cDNAs were synthesized using the PrimeScript™ RT reagent Kit (Takara) from 1 μg RNA following the manufacturer's instructions. The quantitative real-time PCR (qPCR) assays were performed using the TB Green® *Premix Ex Taq*™ kit (Takara) and an ABI 7500 Real-Time PCR Detection System (Applied Biosystems, California, USA). The qPCR thermal profile was as follows: one cycle of initial denaturation at 95 °C for 10 min, followed by 40 cycles at 95 °C for 15 s and 60 °C for 34 s. The primers used are detailed in Supplementary Table S1. Three biological replicates were performed for each RNA sample. The *Arabidopsis Actin2* and sugarcane Glyceraldehyde-3-phosphate dehydrogenase (*GAPDH*) were used as internal reference genes as appropriate. The qPCR data were normalized as relative expression levels by the $2^{-\Delta\Delta Ct}$ method[68].

### Yeast one-hybrid assay
To construct the bait and prey vectors, normal (sL) and mutated (msL) ABRE motifs of the $P_{SCBV-YZ2060}$ promoter (721–810 nt) were amplified by PCR (see primers in Supplementary Table S2). Amplicons were cloned into the pAbAi vector digested with *Hin*dIII and *Xho*I using the In-Fusion HD Cloning Kit. This generated bait vectors sL-AbAi and msL-AbAi for sL ABRE and msL ABRE, respectively. The full-length ORFs of ScbZIP72 and AREB1 were amplified by PCR (see primers in Supplementary Table S2). Amplified fragments were cloned into the *Eco*RI and *Bam*HI sites of pGADT7 using the In-Fusion HD Cloning Kit. This generated prey vectors pGADT7-ScbZIP72 and pGADT7-AREB1 for ScbZIP72 and AREB1, respectively. To examine the interactions of ScbZIP72 and AREB1 with the $P_{SCBV-YZ2060}$ promoter, the yeast one-hybrid (Y1H) assay was performed as described in the manual of the Matchmaker Gold Y1H Library Screening System (Clontech, Dalian, China). The yeast cells of Y1H Gold strain co-transformed with the prey and the bait were grown for three days on SD/-Ura/-Leu medium with and without 500 ng/mL Aureobasidin A (AbA). p53-AbAi/pGADT7-53 and sL-AbAi/pGADT7 were used as positive and negative control, respectively. The experiments were performed independently three times.

### Electrophoretic mobility shift assay
The trimmed ScbZIP72 (870 bp) was cloned into vector pET-28a and transferred into competent *Escherichia coli strain* BL21 (DE3) to produce the His-ScbZIP72 fused protein with 0.6 mM isopropylthio-β-galactiside for 20 h at 18 °C. This His-ScbZIP72 fused protein was purified using the Ni-NTA column based on the His-tag (Lablead, Beijing, China). The sL of biotin-labeled and no label at the 3′-end were used as binding and competitive probes, respectively. The msL of biotin-labeled at the 3′-end was used as the control of binding probe. The EMSA was performed according to the manual of LightShift Chemiluminescent EMSA Kit (Thermo Scientific, Rockford, USA). The oligonucleotide probes are listed in the Supplementary Data 1.

### Statistics and reproducibility
The analysis of variance (One-way ANOVA) was performed for each data set of different treatments or time-points. The mean differences at $P < 0.05$ were conducted using Duncan's multiple range test, and mean values with the same letter are not significantly different. Besides, the Student's *T* test was used for comparison of means differences between treatment and control groups, and $P = 0.05$ and $P = 0.01$ are indicated by one and two asterisks, respectively. All analyses were conducted with the software SPSS 22 Statistic Program. The n number in each experiment represents independent experiments and are labeled in the figure legend. All the experiments were performed at least three times independently.

### Reporting summary
Further information on research design is available in the Nature Portfolio Reporting Summary linked to this article.

### Data availability
The source data used to Figures and Supplementary Figs. are available in the Supplementary Data 2 and 3. Supplementary Fig. S8 and Supplementary Fig. S9 contain the original uncropped EMSA/gel images of Fig. 7c and Supplementary Fig. 5a, respectively. Other data are available from the corresponding authors on reasonable request.

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

## Acknowledgements
This research was supported by the China Agriculture Research System (grant no. CARS-170302) and the National Natural Science Foundation of China (32201735).

## Author contributions
S.J.G. and P.R. designed the experiments and edited the manuscript; S.R.S., X.B.W., J.S.C., M.T.H., H.Y.F., Q.N.W. performed the experiments; S.R.S., X.B.W. and S.J.G. analyzed experimental results; S.R.S., S.J.G. and P.R. wrote the manuscript.

## Competing interests
The authors declare no competing interests.
