## [Peer review file · Communications Biology]

Reviewers' comments:

Reviewer #1 (Remarks to the Author):

Comments for Author

The manuscript presenting isolation of six different promoters from sugarcane bacilliform virus (SCBV) genotypes. The expression of the promoters was monitored using GUS reported gene. Three (PSCBV-YZ2060, PSCBV-TX and PSCBV-CHN2) of the promoters functioned as drought-induced promoters in transgenic Arabidopsis plants. The authors claimed that using mutation approach the PSCBV-YZ2060 promoter is required to confer drought stress and ABA induction. In addition, the transcription factor of ScbZIP72 and AREB1 were bound with ABREs of the PSCBV-YZ2060 and enhanced the expression in response to ABA and PEG6000 treatments. Overall the manuscript presented in a simple and understandable English, and the results were presented in a good manner. However, the authors should clarify following questions before the manuscript is accepted.

1.The 25% PEG6000 treatment was used to induce drought stress, but this chemical is an artificial drought stress. Why were the transgenic sugarcane and Arabidopsis not subjecting the natural drought stress by stop watering?

2.Fig 4B, why the expression of GUS driven by PSCBV-YZ2060 was decreased at 24 h after PEG treatments, but not in ABA (Fig 4A) and also in sugarcane stem (Fig 8B). Please clarify in the manuscript. Furthermore, the symbol of significant different is not consistent, the Fig 4 and 5C using asterisk, but the other using the latter.

3.Fig 2, Fig 6, Fig 8, the pattern of GUS protein activity assay is not match with RT-qPCR analysis. Please clarify why that is happen.

4.The authors suggested that the promoter PSCBV-YZ2060 expressions have tissue specific pattern (line 273-274 and Fig S4). What is the reason behind the suggestion. However, the GUS expression pattern did not show the tissue specific pattern after addition of either PEG or ABA in Fig 8A.

5.There are 55 lines of transgenic sugarcane in line 264, but only 21 lines transgenic lines were analyzed. Please describe the reason.

Reviewer #2 (Remarks to the Author):

The work is related to the identification and full characterization of a novel sugarcane bacilliform virus promoter regulated by transcription factor ScbZIP72 and triggering a drought stress response in plants. The authors previously have been explored alternative promoters from plant viruses with desirable features in plant engineering. The study's major finding is the identification of PSCBV-YZ2060 as an alternative promoter for engineering drought-resistant transgenic crops, particularly sugarcane. The researchers discovered that this promoter acted as an ABA-induced and drought stress promoter in both monocot (sugarcane) and dicot (Arabidopsis) plants. They also found that the sugarcane ScbZIP72 and Arabidopsis AREB1 transcription factors bound to ABREs of the PSCBV-YZ2060 promoter. These findings highlight the potential of PSCBV-YZ2060 as a valuable genetic tool for developing drought-tolerant transgenic crops. The study design and methods employed were suitable for answering the research questions, and the conclusions drawn were well-supported by the presented evidence. Additionally, the use of statistics and the treatment of uncertainties were appropriate.

Dear authors, please revise the manuscript incorporating the following comments and corrections:

In line 122 of the Results section, it should be noted that the 3.0 kb amplified region was selected for the sequence comparison of eight SCBV promoters because it encompasses the RT-RNaseH-coding region, which is widely regarded as the most common taxonomic marker for identifying badnaviral genomic components. This coding region is a standard source to compare the sequence diversity of the badnaviral genomes (Bousalem M., Douzery E.J., and Seal S. (2008). Taxonomy, molecular phylogeny, and evolution of plant reverse transcribing viruses (family Caulimoviridae) inferred from the full-length genome and reverse transcriptase sequences. Archives of Virology, 153(6): p. 1085–1102.

In line 450 of the Material and Methods section, please change https://www.fruitfly.org/seq_tools/promoter.html to https://www.fruitfly.org/seq_tools/promoter.html.

In lines 485-487 of the M&M section: please check the NCBI library because the sequences under accession numbers OL322080-OL322089 are not available.

And, just for curiosity, I would like to know if the authors have determined the number of transgene copies inserted in the transgenic sugarcane lines used in this study.

I don't have any further comments to add. The research work is exceptionally high in quality. I'd like to extend my best wishes to the authors for their significant contribution to the scientific community and professionals involved in sugarcane improvement. Congratulations to all the authors!

Reviewer #3 (Remarks to the Author):

Title of the manuscript: Identification of a novel sugarcane bacilliform virus promoter regulated by transcription factor ScbZIP72 and triggering drought stress response in plants.

In this study, the authors revealed that three promoters cloned from sugarcane bacilliform virus (SCBV) (PSCBV-YZ2060, PSCBV-TX and PSCBV-CHN2) functioned as drought induced promoters in transgenic Arabidopsis plants. They further showed that a putative promoter region 1 (PPR1) and two ABA response elements (ABREs) were required in the promoter PSCBV-YZ2060 to confer drought stress response and ABA induction. They reported that the expression levels of endogenous ScbZIP72 and heterologous GUS were significantly increased in PSCBV-YZ2060:GUS transgenic sugarcane plants after ABA treatment or drought stress. Further, they established that ScbZIP72 from sugarcane and AREB1 from Arabidopsis bound to ABREs element in the promoter PSCBV-YZ2060 in a yeast one-hybrid assay. Authors provided an interesting insight of the Promoter PSCBV-YZ2060 which could alternatively be used for genetic modification of drought-resistant transgenic crops.

The manuscript is interesting and well written; nonetheless, I have listed some clarifications and suggestions that may help the authors to further strengthen the conclusion.

1. In the material and method section, it is important to include detailed manufacturer information for each kit used in the study.
2. In material and methods section/results section, authors need to explain in detail how the transactivation activities of ScbZIP72 and AREB1 were confirmed when yeast cells of Y1H Gold strain co-transformed with the prey and the bait were grown on SD/-Ura/-Leu medium with and without

500 ng/mL Aureobasidin A (AbA). i.e visualization of blue colonies on the medium. The positive control and negative control used in the Y1H assay need to be reported in the methods section.

3. The authors may conduct Luciferase transactivation assay in Arabidopsis protoplast to further establish that ScbZIP72 from sugarcane and AREB1 from Arabidopsis mediated activation on PSCBV-YZ2060.

4. What motivate the authors to choose ScbZIP72 from sugarcane and AREB1 from Arabidopsis out of numerous TFs, considering the fact that a wide range of TFs may bind with the two ABREs of promoter PSCBV-YZ2060.

5. The authors indicated that ScbZIP72 from sugarcane and AREB1 from Arabidopsis could bind specifically to the ABRE regulatory element in the promoter PSCBV-YZ2060 through Y1H assay. Authour may need to ascertain this with strong evidence. Therefore, I suggest the use of EMSA assay to further establish the binding of ScbZIP72 from sugarcane and AREB1 from Arabidopsis to the ABRE regulatory element in the promoter PSCBV-YZ2060

6. Detailed statistical analysis used need to be reported in the result section (more importantly in the figure legend). i.e the student's t test is used to compare the means between two groups, or ANOVA used to compare the means among three or more groups.

Dear reviewers,

We thank your constructive suggestions and comments. We have provided point-by-point responses to the comments below.

Reviewer #1:

Comment 1: The manuscript presenting isolation of six different promoters from sugarcane bacilliform virus (SCBV) genotypes. The expression of the promoters was monitored using GUS reported gene. Three (PSCBV-YZ2060, PSCBV-TX and PSCBV-CHN2) of the promoters functioned as drought-induced promoters in transgenic *Arabidopsis* plants. The authors claimed that using mutation approach the PSCBV-YZ2060 promoter is required to confer drought stress and ABA induction. In addition, the transcription factor of ScbZIP72 and AREB1 were bound with ABREs of the PSCBV-YZ2060 and enhanced the expression in response to ABA and PEG6000 treatments. Overall the manuscript presented in a simple and understandable English, and the results were presented in a good manner. However, the authors should clarify following questions before the manuscript is accepted.

Reply: Thank you for your positive comment.

Comment 2: The 25% PEG6000 treatment was used to induce drought stress, but this chemical is an artificial drought stress. Why were the transgenic sugarcane and *Arabidopsis* not subjecting the natural drought stress by stop watering?

Reply: Soil-drought treatment in transgenic sugarcane and *Arabidopsis* require long time and exhibit certain variables from environmental conditions. The 25% PEG6000 treatment on plants is a common approach for induce drought stress. It is a convenient and time-saving experiment in the lab conditions. This experiment on these transgenic plants under soil-drought treatment might be further investigated in another project.

Comment 3: Fig 4B, why the expression of GUS driven by PSCBV-YZ2060 was decreased at 24 h after PEG treatments, but not in ABA (Fig 4A) and also in sugarcane stem (Fig 8B). Please clarify in the manuscript. Furthermore, the symbol of significant different is not consistent, the Fig 4 and 5C using asterisk, but the other using the latter.

Reply: For the first argument, the expression of GUS driven by PSCBV-YZ2060 was decreased at 24 h after PEG treatments, but not in ABA (Fig 4A & Fig 8B). The reason is that plants treated with 25% PEG6000 after 24 h looked in the status of serious

dehydration and wilting in either sugarcane or *Arabidopsis*, but this appearance was not showed in plants treated by 10 μ M ABA after 24 h.

For the second argument about the symbol of significant difference, the analysis of variance (One-way ANOVA) was performed for each data set of different treatments or time-points. Therefore, the same letter between groups is not significantly difference at $P = 0.05$. However, the Student's T-test was used for comparison of means between two groups (treatment vs. control). We have added these contents in the "Statistical analyses" section and the legend of figures.

Comment 4: Fig 2, Fig 6, Fig 8, the pattern of GUS protein activity assay is not match with RT-qPCR analysis. Please clarify why that is happen.

Reply: This is a very common phenomenon that the slight difference of GUS activities exhibits between protein (determined by ELISA) and mRNA (determined by RT-qPCR) levels because the RT-qPCR is more sensitive than ELISA. However, the overall trend of GUS expression at both protein and mRNA levels is identical in each set experiment.

Comment 5: The authors suggested that the promoter PSCBV-YZ2060 expressions have tissue specific pattern (line 273-274 and Fig S4). What is the reason behind the suggestion. However, the GUS expression pattern did not show the tissue specific pattern after addition of either PEG or ABA in Fig 8A.

Reply: The promoter PSCBV-YZ2060 exhibited the tissue-specific expression in sugarcane plants revealed by GUS protein level (the updated supplemental Figure 4 and the Figure 8B), however, there is no obvious tissue-specific pattern of GUS expression at mRNA level determined by RT-qPCR (Fig. 8A). The reason would be resulted by the different analytical methods and different expression levels (transcript vs. translation).

Reviewer #2 (Remarks to the Author):

Comment 1: The work is related to the identification and full characterization of a novel sugarcane bacilliform virus promoter regulated by transcription factor ScbZIP72 and triggering a drought stress response in plants. The authors previously have been explored alternative promoters from plant viruses with desirable features in plant engineering. The study's major finding is the identification of PSCBV-YZ2060 as an alternative promoter for engineering drought-resistant transgenic crops, particularly sugarcane. The researchers discovered that this promoter acted as an ABA-induced and

drought stress promoter in both monocot (sugarcane) and dicot (*Arabidopsis*) plants. They also found that the sugarcane ScbZIP72 and *Arabidopsis* AREB1 transcription factors bound to ABREs of the PSCBV-YZ2060 promoter. These findings highlight the potential of PSCBV-YZ2060 as a valuable genetic tool for developing drought-tolerant transgenic crops. The study design and methods employed were suitable for answering the research questions, and the conclusions drawn were well-supported by the presented evidence. Additionally, the use of statistics and the treatment of uncertainties were appropriate.

Reply: Thank you for your positive comment.

Comment 2: In line 122 of the Results section, it should be noted that the 3.0 kb amplified region was selected for the sequence comparison of eight SCBV promoters because it encompasses the RT-RNaseH-coding region, which is widely regarded as the most common taxonomic marker for identifying badnaviral genomic components. This coding region is a standard source to compare the sequence diversity of the badnaviral genomes (Bousalem M., Douzery E.J., and Seal S. (2008). Taxonomy, molecular phylogeny, and evolution of plant reverse transcribing viruses (family Caulimoviridae) inferred from the full-length genome and reverse transcriptase sequences. *Archives of Virology*, 153(6): p. 1085–1102.

Reply: As you required, we have performed the phylogenetic analysis of all SCBV isolates from this study and NCBI dataset based on the 3.0-kb amplified fragments. The phylogenetic tree was showed in the updated supplemental Figure 1. Overall, similar topology structure was observed among three phylogenetic trees based on different fragment regions.

Comment 3: In line 450 of the Material and Methods section, please change https://www.fruitfy.org/seq_tools/promoter.html to https://www.fruitfly.org/seq_tools/promoter.html.

Reply: We revised it as you required.

Comment 4: In lines 485-487 of the M&M section: please check the NCBI library because the sequences under accession numbers OL322080-OL322089 are not available.

Reply: These sequences have been released by the NCBI online now.

Comment 5: And, just for curiosity, I would like to know if the authors have determined the number of transgene copies inserted in the transgenic sugarcane lines used in this study.

Reply: We did not check the number of transgene copies inserted in the transgenic sugarcane lines. The effect of transgene copies might be further investigated in another project.

Comment 6: I don't have any further comments to add. The research work is exceptionally high in quality. I'd like to extend my best wishes to the authors for their significant contribution to the scientific community and professionals involved in sugarcane improvement. Congratulations to all the authors!

Reply: Thank you for your nice wishes.

Reviewer #3 (Remarks to the Author):

Comment 1: Title of the manuscript: Identification of a novel sugarcane bacilliform virus promoter regulated by transcription factor ScbZIP72 and triggering drought stress response in plants.

In this study, the authors revealed that three promoters cloned from sugarcane bacilliform virus (SCBV) (PSCBV-YZ2060, PSCBV-TX and PSCBV-CHN2) functioned as drought induced promoters in transgenic Arabidopsis plants. They further showed that a putative promoter region 1 (PPR1) and two ABA response elements (ABREs) were required in the promoter PSCBV-YZ2060 to confer drought stress response and ABA induction. They reported that the expression levels of endogenous ScbZIP72 and heterologous GUS were significantly increased in PSCBV-YZ2060:GUS transgenic sugarcane plants after ABA treatment or drought stress. Further, they established that ScbZIP72 from sugarcane and AREB1 from Arabidopsis bound to ABREs element in the promoter PSCBV-YZ2060 in a yeast one-hybrid assay. Authors provided an interesting insight of the Promoter PSCBV-YZ2060 which could alternatively be used for genetic modification of drought-resistant transgenic crops. The manuscript is interesting and well written; nonetheless, I have listed some clarifications and suggestions that may help the authors to further strengthen the conclusion.

Reply: Thank you for your comments.

Comment 2: In the material and method section, it is important to include detailed manufacturer information for each kit used in the study.

Reply: We added detailed manufacturer information for each kit.

Comment 3: In material and methods section/results section, authors need to explain in detail how the transactivation activities of ScbZIP72 and AREB1 were confirmed when yeast cells of Y1H Gold strain co-transformed with the prey and the bait were grown on SD/-Ura/-Leu medium with and without 500 ng/mL Aureobasidin A (AbA). i.e visualization of blue colonies on the medium. The positive control and negative control used in the Y1H assay need to be reported in the methods section.

Reply: We add these contents in the M&M section as your required.

Comment 4: The authors may conduct Luciferase transactivation assay in Arabidopsis protoplast to further establish that ScbZIP72 from sugarcane and AREB1 from Arabidopsis mediated activation on PSCBV-YZ2060.

Reply: Thank you for your good suggestion. However, the luciferase transactivation assay is unavailable in our lab. Alternatively, we used the EMSA method to check ScbZIP72 *in vitro* binding on PSCBV-YZ2060 promoter. The result was showed in the Figure 7C.

Comment 5: What motivate the authors to choose ScbZIP72 from sugarcane and AREB1 from Arabidopsis out of numerous TFs, considering the fact that a wide range of TFs may bind with the two ABREs of promoter PSCBV-YZ2060.

Reply: Numerous cases revealed that bZIP72 and AREB1 are responsible to drought, so we selected them used in this study.

Comment 6: The authors indicated that ScbZIP72 from sugarcane and AREB1 from Arabidopsis could bind specifically to the ABRE regulatory element in the promoter PSCBV-YZ2060 through Y1H assay. Author may need to ascertain this with strong evidence. Therefore, I suggest the use of EMSA assay to further establish the binding of ScbZIP72 from sugarcane and AREB1 from Arabidopsis to the ABRE regulatory element in the promoter PSCBV-YZ2060

Reply: We performed this experiment as you required, and the result is showed in the Figure 7C. Results between EMSA and Y1H assays were identical.

Comment 7: Detailed statistical analysis used need to be reported in the result section (more importantly in the figure legend). i.e the student's t test is used to compare the means between two groups, or ANOVA used to compare the means among three or more groups.

Reply: We have added these contents in the “Statistical analyses” section and legends of figures.

Reviewers' comments:

Reviewer #1 (Remarks to the Author):

Dear Authors,

Thank you for revision of the manuscript entitled "Identification of novel sugarcane bacilliform virus promoter regulated by transcription factor ScbZIP72 and triggering drought stress response in plants". I recommend the manuscript be published in the Communication Biology and congratulations.

Reviewer #2 (Remarks to the Author):

Dear authors

All my concerns have been addressed in the revised version of the manuscript. I have no more comments. The research work is of very good quality, and the suggestions from the other reviewers have contributed to improving it.

Reviewer #3 (Remarks to the Author):

The manuscript is much better in its present version as the authors have addressed the majority of the issues raised in the manuscript. However, Authors may need to provide more compelling reasons for their choice of bZIP72, considering the fact that various studies have also shown that a large number of other TFs can bind to ABRE regulatory element and also involved in drought stress tolerance. In addition there is no noticeable distinction between ScbZIP72 +probe without competitor and ScbZIP72 +probe with competitor in the EMSA experiment the author claimed to have conducted to further establish the binding of ScbZIP72 from sugarcane to the ABRE regulatory element in the promoter PSCBV-YZ2060 (Fig. 7C). The authors might have to provide an explanation for this.

Bello Babatunde, PhD

Dear reviewers,

We thank you for the constructive suggestions and comments. We have provided point-by-point responses to the comments below.

Reviewer #1 (Remarks to the Author):

Dear Authors,

Thank you for revision of the manuscript entitled "Identification of novel sugarcane bacilliform virus promoter regulated by transcription factor ScbZIP72 and triggering drought stress response in plants". I recommend the manuscript be published in the *Communication Biology* and congratulations.

Reply: Thank you for your positive comment.

Reviewer #2 (Remarks to the Author):

Dear authors

All my concerns have been addressed in the revised version of the manuscript. I have no more comments. The research work is of very good quality, and the suggestions from the other reviewers have contributed to improving it.

Reply: Thank you for your positive comment.

Reviewer #3 (Remarks to the Author):

Comment 1: The manuscript is much better in its present version as the authors have addressed the majority of the issues raised in the manuscript. However, Authors may need to provide more compelling reasons for their choice of bZIP72, considering the fact that various studies have also shown that a large number of other TFs can bind to ABRE regulatory element and also involved in drought stress tolerance.

Reply: Numerous published cases revealed that homologous ScbZIP72 proteins were reported to improve the drought tolerance of maize and rice (Ying, S. et al. 2012. Cloning and characterization of a maize bZIP transcription factor, ZmbZIP72, confers drought and salt tolerance in transgenic *Arabidopsis*, *Planta* 235, 253–266; Lu, G. et al. 2009. Identification of OsbZIP72 as a positive regulator of ABA response and drought tolerance in rice, *Planta* 229, 605-615.). However, The bZIP gene associated with drought tolerance in sugarcane remains unclear, so we decide to select this transcription

factor used in this study.

Comment 2: In addition, there is no noticeable distinction between ScbZIP72 +probe without competitor and ScbZIP72 +probe with competitor in the EMSA experiment the author claimed to have conducted to further establish the binding of ScbZIP72 from sugarcane to the ABRE regulatory element in the promoter PSCBV-YZ2060 (Fig. 7C). The authors might have to provide an explanation for this.

Reply: Thank you for your excellent comment. We have improved the EMSA assay and replaced the image in Fig 7C to address your concerns (please see our reply to comment 1 of the editor).

REVIEWERS' COMMENTS:

Reviewer #3 (Remarks to the Author):

Dear authors,

The manuscript is much better in its present version as all the flaws identified have been addressed. I hereby recommend the manuscript for publication in the Communication Biology.